# Safety and Efficacy of Buprenorphine-Naloxone in Pregnancy: A Systematic Review of the Literature

Alice Ordean [1,2,*] and Meara Tubman-Broeren [3]

1   Department of Family Medicine, St. Joseph's Health Centre, Unity Health Toronto,
    Toronto, ON M6R 1B5, Canada
2   Department of Family and Community Medicine, University of Toronto, Toronto, ON M5G 1V7, Canada
3   Department of Medicine, Faculty of Medicine, University of British Columbia,
    Vancouver, BC V5Z 1M9, Canada
*   Correspondence: alice.ordean@unityhealth.to

**Abstract:** The prevalence of opioid use among pregnant people has been increasing over the past few decades, with a parallel increase in the rate of neonatal abstinence syndrome. Opioid agonist treatment (OAT) including methadone and buprenorphine is the recommended management method for opioid use disorders during pregnancy. Methadone has been extensively studied during pregnancy; however, buprenorphine was introduced in the early 2000s with limited data on the use of different preparations during pregnancy. Buprenorphine-naloxone has been incorporated into routine practice; however, only a few studies have investigated the use of this medication during pregnancy. To determine the safety and efficacy of this medication, we conducted a systematic review of maternal and neonatal outcomes among buprenorphine-naloxone-exposed pregnancies. The primary outcomes of interest were birth parameters, congenital anomalies, and severity of neonatal abstinence syndrome. Secondary maternal outcomes included the OAT dose and substance use at delivery. Seven studies met the inclusion criteria. Buprenorphine-naloxone doses ranged between 8 and 20 mg, and there was an associated reduction of opioid use during pregnancy. There were no significant differences in gestational age at delivery, birth parameters, or prevalence of congenital anomalies between buprenorphine-naloxone-exposed neonates and those exposed to methadone, buprenorphine monotherapy, illicit opioids, or no opioids. In studies comparing buprenorphine-naloxone to methadone, there were reduced rates of neonatal abstinence syndrome requiring pharmacotherapy. These studies demonstrate that buprenorphine-naloxone is a safe and effective opioid agonist treatment for pregnant people with OUD. Further large-scale, prospective data collection is required to confirm these findings. Patients and clinicians may be reassured about the use of buprenorphine-naloxone during pregnancy.

**Keywords:** opioids; pregnancy; buprenorphine-naloxone; neonatal abstinence syndrome





## 1. Introduction

Untreated opioid use disorder (OUD) in pregnancy is associated with significant maternal, fetal, and neonatal risks including fetal growth restriction, preterm labor, and increased perinatal morbidity and mortality [1,2]. Data from the 2020 National Survey on Drug Use and Health indicated that 8.3% of pregnant women in the United States had used illicit drugs in the past month, with 0.4% reporting opioid misuse [3]. The national rate of maternal opioid-related diagnoses in the United States increased from 3.5 in 1000 delivery hospitalizations in 2010 to 8.2 per 1000 in 2017 [4]. Concomitantly, the rates of neonatal abstinence syndrome (NAS) almost doubled in the United States from 4 in 1000 birth hospitalizations in 2010 to 7.3 per 1000 in 2017 [4]. Opioid agonist therapy (OAT) is the recommended treatment for OUD in pregnancy with the proven benefits of decreasing maternal illicit opioid use and improving maternal and neonatal health

outcomes [1,2]. Methadone maintenance treatment has traditionally been considered the standard of care for OUD in pregnancy [2,5–7]. However, in 2010, the first randomized controlled trial of buprenorphine in comparison to methadone in pregnancy demonstrated that buprenorphine was an acceptable alternative with comparable safety and efficacy to methadone [5]. Buprenorphine was also shown to decrease the severity of NAS in comparison to methadone, findings consistent with a larger body of non-randomized studies [5,6].

Buprenorphine is routinely available as a combination product with naloxone, which is intended to act as a deterrent to injection use, due to the risk of precipitated withdrawal. When buprenorphine/naloxone is taken sublingually, naloxone has minimal bioavailability and does not cause any antagonist effect [6,7]. Historically, due to inadequate safety data about the effects of naloxone in pregnancy, buprenorphine monotherapy was recommended instead of the combination product for pregnant people [1,2,7–11]. The need for further research was recommended to establish the safety of buprenorphine/naloxone during pregnancy. More recently, there has been a notable change in the Health Canada approved product monograph for buprenorphine/naloxone (brand name Suboxone®) eliminating pregnancy as a contraindication to its use [2,8]. The goal of this study was to conduct a systematic review of the literature relating to maternal and neonatal safety and efficacy of buprenorphine-naloxone in pregnancy. These findings will serve to update clinical practice guidelines and will impact clinical decision making related to the management of OUD during pregnancy.

## 2. Materials and Methods

### 2.1. Data Sources and Study Selection

The Preferred Reporting Items for Systematic Reviews and Meta-Analyses (PRISMA) (2009) was followed for this systematic review. A comprehensive search strategy was developed in collaboration with an Information Specialist at the University of Toronto. Medline, Embase, and Cochrane Library databases were searched from 1990 until October 2020. Keywords included buprenorphine, naloxone, and pregnancy. Manual reviews of references lists were also performed to ensure that no relevant studies were omitted. The results of this search were first screened for duplicates, and then both authors screened the remaining titles and abstracts for eligibility criteria prior to full-text retrieval. Where decisions were unable to be made from the title and abstract alone, the full paper was retrieved. Disagreements about eligibility were resolved by consensus.

Article were included if they met the following criteria: (a) study included only pregnant people with a history of opioid use or opioid use disorder, (b) buprenorphine-naloxone was used at some point during pregnancy, and (c) primary or secondary outcomes of interest were reported. Only randomized controlled trials and observational cohort or case control studies published in peer-reviewed publications were eligible for inclusion. We excluded expert opinions, editorials, review articles, and guidelines. Articles were also excluded if they were not in the English language. The primary outcomes consisted of gestational age at delivery, birth parameters (birth weight, length, and head circumference), congenital anomalies, and neonatal abstinence syndrome (NAS). Specific NAS measures included neonatal intensive care unit (NICU) admission, prevalence of NAS pharmacotherapy, and duration of hospital stay. The secondary outcomes related to maternal OAT dose and substance use at delivery.

### 2.2. Data Extraction and Analysis

A data extraction spreadsheet was developed and piloted by both authors to ensure inter-rater reliability. Both authors independently extracted data relating to study characteristics, demographics, and outcomes of interest for eligible studies. In cases of disagreement, the full text article was reviewed, and consensus was achieved based on further discussion. Variability in study design and measured outcomes did not allow for meta-analysis of data.

*2.3. Reporting of Study Risk of Bias Assessment*

The Risk of Bias of Non-randomized Studies of Interventions (ROBINS-I) tool was used to assess the risk of bias of the included studies. The ROBINS-I tool comprises seven domains of potential bias, and each domain was assessed as having a low, moderate, serious, or critical risk of bias.

## 3. Results

The literature search identified 168 unique articles, of which 12 full text articles were retrieved for further screening (Figure 1). Seven studies met the inclusion criteria for this systematic review.

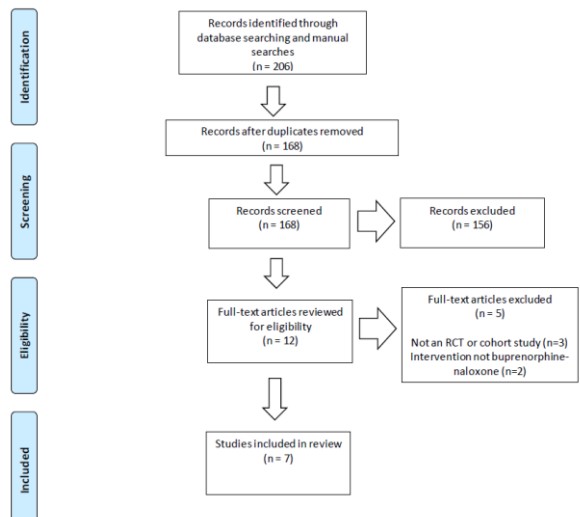

**Figure 1.** PRISMA flow diagram for literature search and selection of articles based on inclusion and exclusion criteria.

*3.1. Study Characteristics*

All included studies were retrospective observational studies involving a total of 302 mother–infant dyads exposed to buprenorphine-naloxone (Table 1). Two studies were performed in outpatient treatment programs in Canada, and the other five studies were conducted in the United States [12–18]. The two Canadian studies compared buprenorphine-naloxone exposure during pregnancy to illicit opioid use or no opioid exposure during pregnancy [17,18]. These two studies originated from the same Northwestern community in Ontario, Canada, and may have included data on the same population of patients, with Jumah et al. extending their study for an additional 6 months in 2015. However, the sample sizes for the buprenorphine-exposed population were significantly different. Dooley et al. reported on 30 buprenorphine-exposed pregnancies, whereas Jumah et al. included 62 buprenorphine-exposed pregnancies. Two studies from the US reported outcomes of single-cohort studies with no comparison group [12,13]. The other three studies compared buprenorphine-naloxone to buprenorphine monotherapy (n = 1) or methadone (n = 2) [14–16]. The majority of studies included participants with any buprenorphine-naloxone exposure in pregnancy, while two studies included only those stabilized on buprenorphine-naloxone at the time of delivery [12–18]. One study excluded patients who switched OAT, including to or from methadone, during pregnancy [14].

**Table 1.** Characteristics of included studies and participant demographics.

| Author, Year | Setting | Number of Participants | Demographics |
|---|---|---|---|
| Debelak et al., 2013 [12] | Outpatient substance abuse and mental health facility Michigan, USA | Bup-nlx group: n = 10 8 initiated pre-pregnancy 2 initiated during first trimester<br><br>No comparison group | Mean age: 26 years White: 90% High school: 60% Single: 90% |
| Dooley et al., 2016 [17] | Outpatient multidisciplinary obstetric program Ontario, Canada | Bup-nlx group: n = 30 30 initiated pre-pregnancy 5 continued for entire pregnancy<br><br>Comparison groups: Other opioids n = 134 No opioids n = 476 | Mean age: 26 years Mean gravidity: 4 Mean parity: 2 *Higher gravidity and parity in bup-nlx group compared to non-exposed group* * |
| Gawronski et al., 2014 [15] | Academic medical center Ohio, USA | Bup-nlx group: n = 58<br><br>Comparison group: Methadone n = 92 | Mean age: 27 years White: 95% High school: 31% *No significant differences between groups* |
| Jumah et al., 2016 [18] | Community-based outpatient Treatment program Ontario, Canada | Bup-nlx group: n = 62 51 initiated pre-pregnancy 11 initiated during pregnancy 48 switched to monoproduct<br><br>Comparison groups: Illicit opioids n = 159 No opioids n = 618 | Mean age: 26 years Mean gravidity: 4 Indigenous: 85% High school: 16% *Higher gravidity in bup-nlx group compared to non-exposed group* * |
| Mullins et al., 2020 [14] | Community-based perinatal substance use disorders program North Carolina, USA | Bup-nlx group: n = 85<br><br>Comparison group: Bup-monoproduct: n = 108 | Mean age: 28 years White: 89% Primiparous: 26% *No significant differences between groups* |
| Nguyen et al., 2018 [13] | Outpatient treatment program For pregnant women with opioid use disorder West Virginia, USA | Bup-nlx group: n = 26<br><br>No comparison group | Mean age: 28 years White: 89% Single: 60% Mean parity: 2 |
| Wiegand et al., 2015 [16] | Residential and outpatient women's substance use disorder treatment program North Carolina, USA | Bup-nlx group: n = 31<br><br>Comparison group: Methadone n = 31 | Mean age: 27 years White: 81% Single: 87% Mean high school educated Primiparous: 26% *No significant differences between groups* |

Bup-nlx: buprenorphine-naloxone; bup-mono: buprenorphine monoproduct; USA: United States of America; * indicates statistical difference ($p < 0.05$)

Maternal demographics were not uniformly reported across studies (Table 1). Participants had a mean age of 26 to 27 years and were predominantly white, with the exception of one study in which the majority were Indigenous [12–18]. Most had some high school education, were predominantly single, and had at least one previous birth [12–16,18]. Studies also reported high rates of concurrent use of tobacco (58–89%), alcohol (~20%), and cannabis (10–61%) among women taking buprenorphine-naloxone [14–18]. Significant demographic differences reported by these studies included higher gravidity and

parity among the buprenorphine-naloxone group and non-exposed individuals in the comparison groups [17,18].

*3.2. Neonatal Outcomes*

3.2.1. Birth Parameters

The mean gestational age at birth for buprenorphine-naloxone-exposed neonates ranged from 37 to 40 weeks [12–18]. One study found a significantly higher mean gestational age at birth in buprenorphine-naloxone-exposed pregnancies than in methadone-exposed pregnancies [16]. Rates of preterm birth (<37 weeks) ranged from 3% to 23%, with no significant differences between comparison groups [12–18].

Birth parameters in buprenorphine-naloxone-exposed neonates, as measured by mean or median birth weight, length, and head circumference, were within the normal range (>3rd percentile) (Table 2) [12–18]. In studies that compared buprenorphine-naloxone to illicit opioids, the buprenorphine-naloxone groups had significantly higher birth weights [17,18]. In studies that compared buprenorphine-naloxone to buprenorphine monotherapy, methadone, or no opioids, there were no significant differences in birth weights between the groups [14–18]. Mean head circumference and length did not differ significantly between buprenorphine-naloxone and comparison groups [14–16].

**Table 2.** Neonatal outcomes in buprenorphine-naloxone-exposed infants.

| Study | Birth Parameters | Neonatal Abstinence Syndrome (NAS) Outcomes | Significant Differences |
|---|---|---|---|
| Debelak [12] | Mean GA: 38 weeks<br>Mean HC: 33 cm<br>Mean length: 46 cm<br>Mean BW: 2816 g | NAS requiring pharmacotherapy: 40%<br>Mean duration of NAS treatment: 7 days<br>Mean hospital LOS: 10 days | No comparison group |
| Dooley [17] | Mean GA: 39 weeks<br>Mean BW: 3569 g | NAS requiring pharmacotherapy: 0% | Higher BW in bup-nlx group compared to other opioids group * |
| Gawronski [15] | Mean GA: 38 weeks<br>Mean HC: 33 cm<br>Mean length: 49 cm<br>Mean BW: 2905 g | NAS requiring pharmacotherapy: 64%<br>Mean duration of NAS treatment: 32 days<br>Mean hospital LOS: 9 days | Lower rate of NAS requiring pharmacotherapy in bup-nlx group than in methadone group * |
| Jumah [18] | Mean GA: 39 weeks<br>Mean BW: 3541 g | NAS requiring pharmacotherapy: 2% | Higher BW in bup-nlx group compared to illicit opioids group * |
| Mullins [14] | Mean GA: 39 weeks<br>Mean HC: 36 cm<br>Mean length: 45 cm<br>Mean BW: 2700 g | NAS requiring pharmacotherapy: 35%<br>Median duration of NAS treatment: 9 days<br>Median hospital LOS: 6 days | No significant differences |
| Nguyen [13] | Mean GA: 37 weeks<br>Mean HC: 35 cm<br>Mean length: 45 cm<br>Mean BW: 2700 g | NAS requiring pharmacotherapy: 19%<br>Mean hospital LOS: 16 days | No comparison group |
| Wiegand [16] | Mean GA: 40 weeks<br>Mean HC: 34 cm<br>Mean length: 50 cm<br>Mean BW: 3175 g | NAS requiring pharmacotherapy: 25%<br>Mean duration of NAS treatment: 11 days<br>Mean hospital LOS: 6 days | Higher GA at delivery, lower rates of NAS requiring pharmacotherapy and shorter LOS in bup-nlx group compared to methadone group |

Mean GA: mean gestational age at delivery; HC: mean head circumference; BW: birth weight; LOS: length of stay; bup-nlx: buprenorphine-naloxone; * indicates statistically significant difference ($p < 0.05$).

3.2.2. Congenital Anomalies

Rates of congenital anomalies in buprenorphine-naloxone-exposed neonates ranged from 3.2–3.5%, with no significant differences as compared to buprenorphine monotherapy, illicit opioids, or no opioids [14,17,18].

3.2.3. Neonatal Abstinence Syndrome

Among neonates exposed to buprenorphine-naloxone, rates of NAS requiring pharmacotherapy ranged widely from 0% to 64% (Table 2) [12–18]. The studies compar-

ing buprenorphine-naloxone to methadone found significantly lower rates of NAS with buprenorphine-naloxone [15,16]. The study comparing buprenorphine-naloxone to buprenorphine monotherapy also found significantly lower rates of NAS requiring pharmacotherapy with buprenorphine-naloxone; however, this difference was not significant when adjusted for other variables such as preterm delivery and dose of buprenorphine at delivery [14]. The studies comparing buprenorphine-naloxone to illicit opioids found no significant differences in the rates of NAS requiring pharmacotherapy [17,18]. Similarly, there were no significant differences in the duration of treatment for NAS or rates of NICU admission [12–18].

### 3.3. Maternal Outcomes

The results from included studies indicated that buprenorphine-naloxone was effective at reducing opioid use by delivery among women with opioid use disorders [12–18] (Table 3). Substance use at delivery was measured by urine drug screening (UDS) in five studies and by self-report confirmed by UDS in one study [12–17]. According to these measures, the rates of substance use at delivery ranged widely from 0% to 55% (Table 4) [12–17]. Specifically, women prescribed buprenorphine-naloxone reported lower rates of illicit opioid use compared to those not using any opioid agonist medication [17,18]. There were conflicting findings about substance use at delivery when buprenorphine-naloxone was compared to methadone use during pregnancy. One study found women who were prescribed buprenorphine-naloxone had higher rates of substance use at delivery than those on methadone maintenance treatment, whereas another study did not show any differences in urine toxicology positivity rates between the two groups [15,16].

**Table 3.** Maternal outcomes at delivery.

| Study | Buprenorphine-Naloxone Group Outcomes | Significant Differences |
|---|---|---|
| Debelak [12] | Bup-nlx dose [1]: 8–16 mg<br>UDS pos for illicit drugs [2]: 0% | No comparison group |
| Dooley [17] | Quit illicit opioid use: 80%<br>Reduced illicit opioid use: 10% | Higher rate of illicit opioid cessation in bup-nlx group compared to illicit opioid group * |
| Gawronski [15] | Mean bup-nlx dose: 20 mg<br>UDS pos for illicit drugs: 47% | Higher rate of pos UDS in bup-nlx group compared to methadone group * |
| Jumah [18] | Mean bup-nlx dose: 8 mg<br>Prenatal opioid use: 18% | Lower rate of daily prenatal opioid use in bup-nlx group compared to illicit opioid group * |
| Mullins [14] | Median bup-nlx dose: 12 mg<br>UDS pos for illicit drugs: 55% | Lower median dose in bup-nlx group compared to bup-monoproduct group * |
| Nguyen [13] | UDS pos for illicit drugs: 35% | No comparison group |
| Wiegand [16] | Mean bup-nlx dose: 14 mg<br>UDS pos for illicit drugs: 20% | No significant differences |

[1] Mean bup-nlx dose: mean buprenorphine-naloxone dose at delivery. [2] UDS pos for illicit drugs: urine drug screen positive for illicit drugs at delivery. * indicates statistical difference ($p < 0.05$).

**Table 4.** Risk of bias assessment for included studies.

| Study | Confound-ing | Selection Bias | Classification of Interventions | Deviations from Intended Interventions | Missing Data | Measure-ment of Outcomes | Selection of Reported Results |
|---|---|---|---|---|---|---|---|
| Debelak et al. [12] | Low | Low | Low | Low | Low | Low | Low |
| Dooley et al. [17] | Moderate | Low | Moderate | Low | Low | Low | Low |
| Gawronski et al. [15] | Moderate | Low | Moderate | Low | Low | Low | Low |
| Jumah et al. [18] | Moderate | Low | Moderate | Low | Low | Low | Low |
| Mullins et al. [14] | Low | Moderate | Low | Low | Low | Low | Low |
| Nguyen et al. [13] | Low | Low | Low | Low | Low | Low | Low |
| Wiegand et al. [16] | Low | Low | Low | Low | Low | Low | Low |

*3.4. Risk of Bias in Included Studies*

A summary of the risk of bias in each domain for included studies is presented in Table 4. Three of the included studies showed a low overall risk of bias based on these domains [12,13,16]. The other four studies were judged to be at low or moderate risk of bias for all domains. The studies by Dooley et al. and Jumah et al. were classified as being at higher risk for confounding and classification bias due to their opioid-exposed comparison group consisting of both women using other forms of OAT and women using illicit opioids [17,18]. Gawronski et al. was also deemed to be at higher risk for confounding and classification of interventions due to the lower compliance rate with buprenorphine-naloxone compared to methadone [15]. Mullins et al. was deemed to be at higher risk for selection bias since the choice of medication was at the discretion of the prescribing physician [14].

**4. Discussion**

Among these heterogeneous studies, the demographic and substance use characteristics of the women included in these cohorts are typical of those presenting for OAT in pregnancy, consisting of women in their late 20s, mostly single, and most with a high school education [7,10]. There were no reports of adverse effects in buprenorphine-naloxone-exposed pregnancies compared to those exposed to buprenorphine monotherapy, methadone, illicit opioids, or no opioids. Birth weight, length, and head circumference as well as gestational age at delivery were not significantly different among neonates exposed to buprenorphine-naloxone [12–18]. In addition, rates of congenital anomalies in buprenorphine-naloxone-exposed neonates were comparable to expected rates in the general population [19,20].

The findings of significantly lower rates of NAS requiring pharmacotherapy and shorter duration of hospital stay in buprenorphine-naloxone groups are consistent with existing evidence of reduced severity of NAS in neonates exposed to buprenorphine compared to those exposed to methadone [5,6]. The wide range of rates of pharmacotherapy for the management of NAS in buprenorphine-naloxone-exposed neonates may be explained by differences between studies in NAS assessment and management, including the threshold to initiate pharmacotherapy for NAS, rooming-in policies, and levels of antenatal opioid exposure. The practice of rooming-in has been shown to decrease the need for pharmacotherapy for NAS; however, only one study explicitly stated whether a rooming-in policy was in place [18,21]. Studies that reported low rates of NAS pharmacotherapy promoted low-dose OAT protocols and opioid tapering prior to delivery [17,18]. In both of these studies, the extremely low rates of NAS likely reflect the neonates' minimal exposure

to opioids prior to delivery, as opposed to any characteristic of buprenorphine-naloxone. However, maintenance treatment with OAT is recommended over medical detoxification or rapid tapering off of OAT due to adverse outcomes, such as high relapse rates or return to use and maternal overdose [1,7].

This review also found that buprenorphine-naloxone was effective in reducing illicit opioid use as demonstrated by lower rates of substance use at delivery [17,18]. In one study comparing buprenorphine-naloxone to methadone, buprenorphine-naloxone was associated with a higher positive UDS rate at delivery, which is likely attributable to reduced adherence with buprenorphine/naloxone (86%) dosing compared to methadone (99%) [15]. While early studies showed buprenorphine to be less efficacious than methadone, subsequent studies have consistently found the efficacy to be equivalent when rapid induction and sufficient dosage are used [22]. This is in keeping with the other included study comparing buprenorphine-naloxone to methadone, which found no statistically significant difference in rates of substance use at delivery between the two groups [16].

Our results related to the use of buprenorphine/naloxone during pregnancy are similar to those from another recent publication [23]. Link et al. conducted a systematic review and meta-analysis that included only five studies of 291 buprenorphine-exposed pregnancies compared to other opioid exposures, mainly methadone and buprenorphine. The articles meeting inclusion criteria varied from those in our systematic review. Link et al. excluded any studies without opioid-exposed comparison group(s) and studies where they could not determine OAT use. Their selection process facilitated the ability to conduct meta-analyses for neonatal birth and NAS-related outcomes. Since the goal of the review by Link et al. was primarily related to neonatal outcomes, maternal demographics and maternal outcomes such as substance use in addition to OAT and maternal OAT dose at delivery were not adequately addressed. These maternal parameters are important details when determining the applicability of findings to a particular patient population.

Similar to our conclusions, Link et al. suggested that buprenorphine-naloxone use during pregnancy resulted in similar pregnancy outcomes compared to women on other forms of OAT based on their included studies. No serious adverse maternal or neonatal outcomes were associated with the use of buprenorphine-naloxone during pregnancy. The only significant finding based on their meta-analysis was that neonates exposed to buprenorphine were less likely to require treatment for NAS compared to methadone-exposed neonates. The authors also acknowledged the limitations in terms of the number and quality of the studies regarding the use of buprenorphine-naloxone in pregnancy.

*Limitations*

This systematic review is limited by a small overall population of ~300 buprenorphine-naloxone-exposed dyads with minimal racial diversity. The studies were heterogeneous in terms of timing and duration of buprenorphine-naloxone exposure in pregnancy, reported outcomes, and NAS protocols. Furthermore, all studies consisted of retrospective cohorts focused on short-term outcomes, with no longitudinal and developmental data available. The lack of prospective research, including randomization into exposure groups, is another limitation of the current data. The most significant concern identified was the lack of control for confounding variables in study analyses. These variables included higher rates of smoking in buprenorphine-naloxone groups [14–18]. These factors would be expected to potentially confound results by increasing adverse outcomes in buprenorphine-naloxone groups; however, some studies did attempt to control for the presence of polysubstance use as a confounding variable in their analysis, and poorer outcomes were not seen in these results.

## 5. Conclusions

In this systematic review of the available literature, the results from included studies consistently showed no evidence of maternal or neonatal safety concerns with the use of buprenorphine-naloxone in pregnancy. Buprenorphine-naloxone was reported to be

associated with reduced substance use during pregnancy, as well as reduced severity of NAS when compared to methadone. Clinicians should counsel pregnant people about the benefits and risks of initiating or continuing buprenorphine-naloxone as an alternative for the management of opioid use disorders during pregnancy.

**Author Contributions:** Conceptualization, A.O. and M.T.-B.; methodology, A.O.; software, A.O. and M.T.-B.; validation, A.O. and M.T.-B.; formal analysis, A.O. and M.T.-B.; investigation, A.O. and M.T.-B.; resources, A.O. and M.T.-B.; data curation, A.O. and M.T.-B.; writing—original draft preparation, A.O. and M.T.-B.; writing—review and editing, A.O. and M.T.-B.; visualization, A.O. and M.T.-B.; supervision, A.O.; project administration, A.O.; funding acquisition, A.O. All authors have read and agreed to the published version of the manuscript.

**Funding:** This research received no external funding.

**Institutional Review Board Statement:** Not applicable.

**Informed Consent Statement:** Not applicable.

**Data Availability Statement:** No new data were created or analyzed in this study. Data sharing is not applicable to this article.

**Acknowledgments:** We want to thank Naz Torabi, Unity Health Information Specialist, for their assistance with the search strategy.

**Conflicts of Interest:** The authors declare no conflict of interest. Ordean receives salary support for academic work from the Department of Family Medicine at St. Joseph's Health Centre and has also received honoraria for presentations and the development of educational materials relating to substance use in pregnancy. Tubman-Broeren has no conflicts of interest to disclose.

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
