# Peer review of "Safety and Efficacy of Buprenorphine-Naloxone in Pregnancy: A Systematic Review of the Literature"

_pathophysiology, doi:10.3390/pathophysiology30010004_

Round 1

Reviewer 1 Report

I appreciate the authors' time and effort in conducting this review. Generally, I am concerned that this could be a duplication of another already published systematic review with similar objectives: https://doi.org/10.1016/j.ajogmf.2020.100179

Overall, the current systematic review in its current form could be more beneficial to the existing literature if revised to complement the previously published systematic review by Link et al. It would be pertinent to review the similarities and differences between the current review and the previously published review by Link et al. This could be helpful for the authors to identify and redirect the focus to highlight specific characteristics of these articles. For example, birth parameters and mean dose of buprenorphine/naloxone at delivery was not explicitly reported in the review by Link et al. Please see below a few suggestions on how the article could be improved, but again, the major revision would be to revisit how this article could stand out against other similar systematic reviews that already exist in the literature. 

1. Reference 2 link is not functional. I believe this is the correct reference to be used: https://www.bccsu.ca/wp-content/uploads/2018/06/OUD-Pregnancy.pdf

2. Reference 3 link is not functional. This may be the correct reference: https://www.samhsa.gov/data/report/2020-nsduh-detailed-tables

If this reference is used in the final version of this manuscript, I would recommend updating the references to cite the specific table that the data are from, e.g., "2020 NSDUH Detailed Tables: Table 6.20B – Types of Illicit Drug, Tobacco Product, and Alcohol Use in Past Month: Among Females Aged 15 to 44; by Pregnancy Status, Percentages, 2019 and 2020." 

Otherwise, it could be more impactful to replace this with some statistics on opioid-related overdose rates. Here are some data from NIDA: https://nida.nih.gov/research-topics/trends-statistics/overdose-death-rates

3. Reference 4 pertains only the hospital deliveries in the United States, so I would state or clarify this. 

4. Lines 35-37: "The rate of 35 maternal opioid-related diagnoses increased from 3.5 in 1000 delivery hospitalizations in 36 2010, to 8.3 per 1000 in 2017 [4]." - 8.3 should be revised to 8.2. 

5. Lines 37-39: "The prevalence of opioid use in pregnancy is also re-37 flected in the rates of neonatal abstinence syndrome (NAS) which almost doubled in the 38 United States from 4 in 1000 birth hospitalizations in 2010, to 7.3 per 1000 in 2017 [4]." - NAS can also be associated with non-opioid use, so I would revise this sentence. 

6. Lines 48-50: "Buprenorphine is routinely available as a combination product with naloxone, which is intended to act as a deterrent to injection use, due to the risk of precipitated withdrawal, but naloxone has minimal bioavaialability when consumed orally or sublingually [6,7]." - would rephrase or split into two sentences to be clearer, and fix typo for "bioavailability." For example: "Buprenorphine is commonly available as a combination product with naloxone. This combination is intended to act as a deterrent to injection use due to the risk of precipitated withdrawal, but it can safely be used orally or sublingually since naloxone has minimal bioavailability when consumed by mouth [6,7].

7. Lines 53-55: Would specify that the product monograph in reference is related to a Canadian product label. Here is an excerpt from the reference used: "Parallel with evidence supporting the safety and effectiveness of buprenorphine/naloxone, pregnancy was recently removed as a contraindication in the product monograph of the Health Canada-approved buprenorphine/naloxone (brand name Suboxone®)." 

8. Lines 55-57: "The goal of this study was to conduct a systematic review of the literature relating to maternal and neonatal safety and efficacy of buprenorphine-naloxone in pregnancy compared to other opioid agonist treatments." Based on Table 1, it doesn't look like there were any studies that included a comparison to other opioid agonist treatments for OUD. Did the authors mean to say "opioid agonist exposures"? If the comparison group is a key part of the inclusion criteria, it is not clear how Debelak 2013 and Nguyen 2018 made it to the final included articles. Please consider clarifying. 

9. Table 1: in the row for Jumah et al. 2016 and column for Number of participants, "Bup=nlx" should be "Bup-nlx" 

10. Line 117-118: "There were high rates of concurrent use of tobacco (58%-89%), alcohol (~20%), and cannabis (10%-61%) among women taking buprenorphine-naloxone [14-18]." - is this represented somewhere in the tables? If not, suggest adding something along the lines of "(data not reported)" to clarify. 

11. Line 118-121: "Significant demographic differences between those in buprenorphine-naloxone group and comparison groups included higher gravidity and parity than non-exposed individuals [17,18]." - Need to clarify that "significance" was determined by the individual studies, not by the current systematic review. 

12. All tables: It would be helpful to add the reference numbers in brackets next to the Author names in the Study column. 

13. Lines 191-194: "Studies that reported low rates of NAS pharmacotherapy 191 promoted low dose OAT protocols and opioid tapering prior to delivery [17,18]. In both of these studies, the extremely low rates of NAS likely reflect the neonates’ minimal exposure to opioids prior to delivery, as opposed to any characteristic of buprenorphine naloxone." - It would be important to emphasize that rapid tapering is not recommended during pregnancy due to the high risk of overdose. See one example from this recent publication: https://jamanetwork.com/journals/jamanetworkopen/fullarticle/2791566

14. Lines 196-197: "This review also found that buprenorphine-naloxone was effective in reducing illicit opioid use as demonstrated by lower rates of substance use at delivery [17,18]." Is this overstating the findings from Table 3? 

15. Lines 197-199: "In one study comparing buprenorphine-naloxone to methadone, buprenorphine-naloxone was associated with a higher positive UDS rate at delivery which is likely attributed to its pharmacological properties [15]." - this sentence is not clear, please clarify. 

16. Lines 214-215: "These variables included higher rates of smoking, and hepatitis C in buprenorphine-naloxone groups." - Would remove the comma after "smoking" and also clarify which studies included in the review had the higher rates of smoking and/or hepatitis C. 

17. Lines 221-222: "Buprenorphine-naloxone was found to an effective OAT during pregnancy 221 leading to reduced substance use." - how are the authors defining "effective" and again, is "leading to reduced substance use" overstating the findings from Table 3? Are there other characteristics of people who use buprenorphine/naloxone that might lead to less substance use, or is it the specific type of OUD treatment that is contributing to the reduced rate of use? I would clarify this. 

18. Lines 225-226: "clinicians should be reassured that buprenorphine-naloxone is a safe alternative for the management of opioid use disorders during pregnancy." - this conclusion is only based on seven studies with a total of just over 300 dyads. Perhaps this statement could be revised to say how this review can help support clinicians in patient-centered decision making for treatment of OUD. 

19. Please ensure the PRISMA 2020 Checklist for abstracts and manuscripts has been followed and shared as a supplement if required by the journal. 

Author Response

Thank you for your suggestions to improve our manuscript.   I have included a comparison to the systematic review by Link et al. in the discussion section to explain the similarities and differences between our studies.  

In addition, I have made the required revisions in the manuscript.  Please also see my responses below to the itemized list of revisions.

  1. Reference link updated
  2. Reference link updated
  3. Reference updated to 2020 NSDUH table 6.20B  The issue at hand is to show that bup/nlx can be used during pregnancy and as such, additional statistics on overdose are not needed in the introduction.
  4. Clarified that this data relates to US hospital deliveries.
  5. Number revised as suggested
  6. Sentence split into several sentences to clarify point.
  7. Sentence updated to acknowledge that this statement relates to Health Canada approved formulation.
  8. Clarified goal of the study to acknowledge that we included all studies with pregnant people exposed to bup/nlx to increase our sample size.
  9. Revised accordingly.
  10.  We extracted data on other substance use for all studies; however, we decided no to include these details in maternal demographics table in order to reduce clutter.  If recommended, I can expand table 1 and add prevalence rates from reach study.  As an alternative, we included the prevalence rates as ranges in the paragraph.
  11. Statement clarified.
  12. Tables updated by adding in reference numbers.
  13. Statement added and referenced about medical detoxification with OAT not preferred to OAT maintenance.
  14. I do not think that we are overstating findings.  It is important to highlight the maternal benefits of bup/nlx beyond focusing on neonatal benefits.
  15. Sentence clarified.
  16. References added and statement revised.
  17. Sentence revised to reduce overstatement of findings.
  18. Conclusion revised to indicate that these findings can be used by clinicians in their counselling approach.
  19. We will update our PRISMA checklist to the 2020 version and will have it available, if required by the journal.

Reviewer 2 Report

The objective of this paper was to summarize the literature on the safety and efficacy of buprenorphine/naloxone therapy in pregnancy. The authors identify 7 studies that met inclusion criteria and describe results for the outcomes of gestational age at delivery, birth weight, birth length, head circumference, congenital anomalies, neonatal abstinence syndrome, maternal OAT dose and substance use at delivery. The paper is well written, however some clarification is needed on a few points outlined below.

1.     Introduction: the authors note recent results from clinical trials involving buprenorphine/naloxone (line 53) however these do not appear to be cited. These citations should be added.

2.     The inclusion criteria should be clarified:

a.     The first criteria states “study included only pregnancy people with opioid use disorder”, however the Jumah and Dooley papers include comparison groups with no opioid exposure (and no mention of opioid use disorder, so presumably are without).

b.     The introduction states that the study goal was to compare buprenorphine/naloxone to other OAT in pregnancy (line 57), however the composition of the comparison group was not an inclusion criterion and some studies have no comparison group or non-OAT comparison groups. This should be clarified.

3.     The inclusion criteria did not specify study type, however only observational studies were identified. The authors note clinical trials evaluating buprenorphine/naloxone in the introduction. Did these not meet inclusion criteria?

4.     The studies by Dooley and Jumah appear to be in the same population, with a few additional months included in the Jumah paper. This should be noted in the review and also impacts the calculation of 302 exposed patients.

5.     Another systematic review in this area was published in 2020: Link et al. Buprenorphine-naloxone use in pregnancy: a systematic review and metaanalysis. AJOG MFM. While many of the included studies are the same, that review also includes this paper: Nechanska et al. Neonatal outcomes after fetal exposure to methadone and buprenorphine: national registry studies from the Czech Republic and Norway. Addiction, 2017. Did this study not meet inclusion criteria?

a.     Given the slightly different study inclusion, a comparison of conclusions to those made by the Link review would be a useful addition to the discussion.

6.     Confounding is only addressed in the discussion to note that there was a lack of control for confounding variables in the included studies. Confounding control should be included in the study details, in one of the tables. This is essential information for interpreting the results summarized throughout the paper.

a.     The authors note that higher rates of smoking and hepatitis C would be expected to increase adverse outcomes in the buprenorphine/naloxone group, but conclude this was not seen in the results. This discounts the possibility that with proper adjustment, some outcomes for buprenorphine/naloxone may actually have lower risk of neonatal and maternal outcomes. The authors should not use the crude results to speculate that confounding is not an issue.

Author Response

Thank you for your helpful comments for our systematic review.  The following revisions were made to our manuscript in response to your suggestions.

  1. Introduction: Statement was revised by deleting reference to recent clinical trials to reduce confusion.  The section now reads as follows:Historically, due to inadequate safety data about the effects of naloxone in pregnancy, buprenorphine monotherapy was recommended instead of the combination product in pregnant people [1,2,7-11].  The need for further research was recommended to establish the safety of buprenorphine/naloxone during pregnancy.  More recently, there has been a notable change in the product monograph for buprenorphine/naloxone eliminating pregnancy as a contraindication to its use [2,8].  
  2. I have clarified the inclusion criteria as follows:  
    1. Criteria changed to "a) study included only pregnant people with history of opioid use or opioid use disorder"
    2. Introduction section was updated to reflect the fact that we included any study that provided data on buprenorphine/naloxone during pregnancy. Sentence reads now as follows:  "The goal of this study was to conduct a systematic review of the literature relating to maternal and neonatal safety and efficacy of buprenorphine-naloxone in pregnancy."
  3. Wording clarified in introduction to reduce confusion. Clinical trials was used synonymously with studies so all publications meeting inclusion criteria were included in our systematic review.  Our search only located observational studies since no RCTs have been conducted comparing bup/nlx to other OAT or opioids during pregnancy.
  4. I have added a statement to acknowledge the possibility of overlap between these 2 studies.  However, since they have different sample sizes, there may be only be a partial overlap in patient populations included in these 2 studies.
  5. a. Nechanska et al. was not located by our search strategy.  Based on reviewing the aim of this study, it does not meet our inclusion criteria since the focus of this study was to compare neonatal outcomes comparing buprenorphine to methadone.                                                                      b. I have read the systematic review by Link and I have added a paragraph to our discussion explaining the differences between our 2 studies and the similar conclusions drawn despite these differences.
  6. Confounding was also mentioned in the maternal demographics section of our manuscript where the presence of other substance use and prevalence ranges were provided.  An additional clarifying statement about how polysubstance was adjusted for in some studies was added to the limitations paragraph as well.

I hope these clarifications address the points as outlined.  Thank you for your insightful comments.

Round 2

Reviewer 1 Report

Thank you for the responses. With the revisions, the manuscript has improved and is a nice addition to the existing literature. A few additional revisions are listed below for your consideration. 

1. Line 214: "By these measures, rates 213 of substance use at delivery ranged widely from 0% to 55% (Table 3)[12-17]." - Please review - it looks like the Table reference here should be 4. 

2. Lines 251-253: "However, maintenance treatment with OAT is recommended over medical detoxification due to high relapse rates and associated risk of maternal overdose with tapering off OAT [1,7]." - this statement could be revised - the reference previously shared with the authors was referring to RAPID tapering of OAT and the association with overdose. Perhaps this statement could say: "However, maintenance treatment with OAT is recommended over medically supervised withdrawal or rapid tapering off OAT due to adverse outcomes, such as return to use and overdose, associated with withdrawal symptoms during pregnancy." 

3. Lines 255-258: "In one study comparing buprenorphine-naloxone to methadone, buprenorphine-naloxone was associated with a higher positive UDS rate at delivery which is likely attributed to buprenorphine’s partial agonist effects [15]." - perhaps the intention of this sentence should be clarified. Clinically, it would not make sense that a patient would more likely have a positive UDS just because they are taking a partial opioid agonist such as buprenorphine. 

4. Lines 267-270: "The articles meeting inclusion criteria varied from those in our systematic review specifically, Link et al. excluded any studies without opioid-exposed comparison group(s) and studies where they could not determine OAT use." - this is a run-on sentence, please revise. 

Author Response

Thank you for your responses to the first revisions.

I have reviewed your comments and made additional changes as suggested.

  1. I have corrected this reference to Table 4.
  2. I have revised this statement to clarify the intention of this sentence.
  3. I have changed this explanation.  The difference in positive UDS results is attributable to the difference in adherence to the medications.
  4. I have revised the sentence structure.  I have also added a few additional explanations about the differences between the 2 reviews.

Reviewer 2 Report

Thanks to the authors for responding to my comments. No additional comments. 

Author Response

Thank you for your secondary review.

Spell check will be done on final document to ensure all errors are fixed.